# Rethinking Logic in AI: A Novel Benchmark Inspired by Polynomial Analogue of Gandy's Fixed Point Theorem

## Abstract

This paper introduces a novel benchmark for evaluating the logical reasoning capabilities of Large Language Models (LLMs), grounded in the polynomial analogue of Gandy's classical fixed point theorem. Since this theorem can be used to describe the P-complete HornSAT problem, and our benchmark is based on this theorem, our benchmark thus covers all problems from class P and shows that serious problems have already arisen in this class, not to mention those benchmarks whose complexity classes are NP-complete and NP-hard. Drawing on concepts from mathematical logic, we design a parameterized set of recursively definable problems where the objective is for LLMs to predict whether a problem belongs to an inductively definable set of polynomial complexity. By varying the parameters, we generate problem instances of differing complexity. Our experiments reveal that current state-of-the-art LLMs with zero-shots promts fail to reliably solve even the most straightforward cases despite an effective deterministic algorithm existing. Even advanced models like GPT-4 exhibit significant biases in solving benchmark problems. These findings highlight the limitations of modern LLMs as code interpreters, even in basic scenarios, and underscore the necessity for hybrid LLM/interpreter systems. Furthermore, they emphasize the importance of developing quantitative tests for reasoning, given the increasing reliance on LLM-based systems in decision-making applications.

## 1 Introduction

### 1.1 Motivation

The question of whether large language models (LLMs) can think has been a subject of intense debate for many years (Mirzadeht et al., 2024), and it is only getting hotter as large language models solve more and more problems. To address this question, researchers are designing increasingly complex tests and challenges for LLMs, which allow us to assess the cognitive capabilities of these models more precisely. From a logical perspective, a key component of any intelligent system being able to think is the ability to reason and construct proofs from basic axioms, as well as the ability to work with recursive definitions (Goncharov & Nechesov, 2021b) If we look at the problems from the other side, all problems can be divided into complexity classes P, NP-complete, NP-hard, etc. Many benchmarks try to cover all of these classes at once(Lizhou Fan et al., 2024), but if LLMs cannot solve problems from class P well, then it is pointless to try to study questions in the more complex classes NP-complete and NP-hard. These findings are supported by the results obtained from most studies that aim to tackle problems spanning across different complexity levels simultaneously. When we approach problems from the perspective of formalization, the majority of them can be articulated through non-recursive descriptions with first-order logic formulas. However, when a problem permits a recursive representation, its formulation in logical terms becomes significantly more compact. It is convenient to work with such recursive descriptions of problems and send them to the input of LLMs together with the objects for which these problems need to be solved.

In this paper, we propose to focus on a unique combination of problems that lie in the class P and at the same time admit a recursive description. If LLMs cannot solve these problems, then most often it is pointless to move on to more complex problems from the NP-complete and NP-hard

classes, since their complexity becomes higher and at the same time most of them also admit a recursive description. To navigate through recursive structures while remaining within the confines of class P, the PAG theorem (Appendix A) and FPAG theorem (Nechesov & Goncharov, 2024) prove invaluable, enabling us to define sets and functions that can be described recursively with polynomial complexity. In this investigation, we focus on the PAG theorem, demonstrating that even within this framework, LLMs encounter significant difficulties in determining whether an object belongs to an inductively defined collection of entities. The uniqueness of the benchmark is achieved due to its unique properties - in essence, we are trying to solve recursively defined problems, but at the same time do not jump out of the class P.

The HornSAT problem (Adebayo et al., 2022), which lies in the class P and is a simplification of the NP-complete SAT problem, also falls under the recursive description. The HornSAT problem can be solved using a modified PAG theorem, where we can to transfer the main concepts of the PAG theorem from nested lists to the set of horn formulas, and also had to abandon generating families in favor of the set of horn formulas. This was achieved by redefining the function $\gamma$, which now takes some horn formula as input and produces another shorter horn formula according to a recursive P-computable algorithm for Horn satisfiability (hor). The operator $\Gamma^{HF}$ acts on the set of all subsets of horn formulas in the set of all subsets of horn formulas. In this case, the smallest fixed point $\Gamma^*$ of the operator $\Gamma^{HF}$ is P-computable. Without loss of generality, we can assume that the HornSAT problem is a special case of a modified polynomial analogue of Gandy's theorem. A similar approach can be used to describe the 2-SAT problem using a modified PAG theorem. Since the 2-SAT problem reduces to a 2N×2N adjacency matrix, a transitive closure is iteratively constructed for it using a modified version of the Floyd-Warshall algorithm. (2sa). Due to the fact that the HornSAT problem is p-complete, we can assume that our benchmark based on the PAG theorem covers all problems from the class P. This is its uniqueness and universality.

Large language models have a transformative effect on various domains of science, including language processing and understanding (Ouyang et al., 2022), music and speech processing (Agostinelli et al., 2023), smart cities and digital twins of complex objects, and even drug and protein design (Madani et al., 2023). These models, such as GPT-4 (Achiam et al., 2023), Claude (cla) and their open-source competitors like Llama (Dubey et al., 2024) and Mistral (Jiang et al., 2023), have demonstrated remarkable capabilities in tasks ranging from simple text generation to complex mathematical problem-solving. The remarkable advancements of LLMs were enabled by the abundance of unlabelled text data (Penedo et al., 2024) and the effective unsupervised training at scale (Kaplan et al., 2020).

Despite these successes, there are notable limitations in the current methodologies for evaluating the reasoning capabilities of LLMs. Standardized benchmarks (Hendrycks et al., 2020; Yue et al., 2024; Zheng et al., 2023) have shown that LLMs excel in few-shot and zero-shot tasks, often outperforming previous state-of-the-art models by large margins. However, these benchmarks may not fully capture the models' ability to generalize and reason logically, as evidenced by various studies highlighting often unexpected failures Nezhurina et al. (2024) that contradict the claimed strong capabilities.

Addressing these limitations is crucial for applying LLMs in real-world scenarios, where accurate, logical reasoning is essential. For instance, in fields like healthcare, legal services, and automated decision-making, the reliability of AI models can have significant implications for outcomes and trustworthiness. Therefore, there is an urgent need for new evaluation frameworks Gao et al. (2023); Guha et al. (2024); Wang (2024) that can more accurately assess the logical capabilities of LLMs in different settings.

In response to this need, we introduce a new benchmark designed to evaluate the reasoning abilities of LLMs in recursive problems. Our approach involves generating [object, condition] pairs, where objects are nested lists of elements (for example, numbers) and conditions are expressed with recursive functions. The task of the LLM is to determine whether the object satisfies all conditions. This setup, expressed in Python language, leverages the extension of fixed-point Gandy's theorem (Goncharov & Nechesov, 2021b), ensuring that the check can be performed efficiently at most quadratic times relative to the object's size. Despite an efficient verification algorithm, our findings indicate that all tested LLMs fail to solve even simple instances of these problems, highlighting a significant gap in their logical reasoning capabilities. The source code of the benchmark is available at `https://anonymous.4open.science/r/logic-lm-0B9C`.

## 2 RELATED WORK

As the capabilities of large language models themselves increase, so does the number of various benchmarks that test their capabilities in various areas including the ability to solve mathematical and logical problems. Most often, the problems that arise when testing LLMs can be divided into complexity classes: P, NP-complete, and NP-hard.

In our research, we primarily focus on problems from the P class, as we rely on the PAG (Goncharov & Nechesov, 2021b) and FPAG theorems (Nechesov & Goncharov, 2024) the powerful tools for constructing polynomially computable sets of inductively defined objects and also for defining recursive functions of polynomial complexity. This allows us to see how the larger language models handle these constructions.

The complexity arises from LLMs' need to effectively handle recursive operations that inevitably emerge when tackling such problems. Recursive definitions serve as a key factor in increasing complexity. By carefully managing recursion, we can either maintain operations within the P class or transition to NP-complete or NP-hard classes.

As our research shows, even within the complexity class P, there exist tasks that pose a challenge for large language models, suggesting that higher-level classes such as NP-complete and NP-hard become simply unsolvable in most cases. It is futile to engage in experimentation with issues in these domains unless one possesses a comprehensive understanding of the root causes for tasks from class P.

In the work under consideration Lizhou Fan et al. (2024), attention is directed towards tasks spanning various complexity levels, encompassing P, NP-complete, and NP-hard classes. The research aims to comprehensively explore these diverse complexity classes, rather than focusing narrowly on a specific aspect. However as we noted earlier in the context of this work, where researchers try to solve problems belonging to the NP-complete and NP-hard classes without having fully comprehended the intricacies of problems within the class P.

In another work (Rishi Hazra et al., 2024) the solution of the 3-SAT problem using LLM occurred with a small number of free variables (no more than 10). That is a 3-SAT problem in which it is possible to find a solution manually by enumeration. It turned out that even with such a small number of parameters, LLMs cope poorly with them. As expected, with an increase in parameters, their capabilities sharply decrease.

If we look at benchmarks that have made a lot of noise in the LLM community, then of course this is the Alice problem. (Nezhurina et al., 2024) This problem can be solved by an 8-year-old child, but most of the leading LLMs could not do it. If we want to build trusted AI algorithms based on LLMs, then it is very important to start with the simplest problems to understand where they make mistakes and where they do not. The most acceptable complexity class for these problems is in class P. That is why it is so important to study this complexity class in as much detail as possible, which is what we do in this work.

## 3 BACKGROUND

The idea of our benchmark is to use a parametric class of problems that can be efficiently generated and solved, yet the complexity of individual problem instances can be tuned. We revert to the Polynomial Analogue of Gandy's Fixed Point Theorem (Goncharov & Nechesov, 2021b), or PAG-theorem, to define such a class. In this section, we describe how to define this class of problems in simple terms and refer the readers interested in stricter descriptions to Appendix A.

Let us introduce notation for a class of polynomially solvable problems $\mathcal{F}$. Each problem consists of a pair $(P, x)$, where $P \in \mathcal{F}$ is a recursively defined boolean function (a predicate), and $x$ is a candidate object. The objective is to determine whether or not $x$ belongs to $P$, e.g. if

$$P(x) == 1 \tag{1}$$

here 1 in Eq. 1 means logical `True`, and 0 means `False`, the symbol $==$ means a comparison operation.

The objects $x \in \mathcal{X}$ are arbitrarily nested lists of elements $e$. To be concrete, in this work, we consider lists of natural numbers $e \in \mathcal{N}$, but any elements can be used. We inductively denote a nested list $x$ as follows:

$$x = \langle x_0, \ldots, x_N \rangle$$
$$x_k = e_k \quad \text{or} \quad \langle x_{k,0}, \ldots, x_{k,M} \rangle, \quad k \in [0, \ldots, N] \tag{2}$$

We use composite indices of length $d$ to describe elements $x$ at the $d$-th level of nesting.

The boolean functions in $\mathcal{F}$ must adhere to certain structures imposed by the PAG theorem. Consider a set of predicates $\{P_0(x), \ldots, P_I(x)\}$. As objects $x$ are lists, the structure of $P \in \mathcal{F}$ will depend on how it is applied to individual elements of the list. In this work we consider boolean functions on lists $x = \langle x_0, \ldots, x_N \rangle$ of the following form:

$$\Phi_j(x_0, \ldots, x_N) = Q_{j,0}(x_0) * \cdots * Q_{j,N}(x_N)$$
$$Q_{j,i}(x_i) = \begin{cases} P_{j,i}(x_i) \\ (x_i == e_i) \end{cases} \tag{3}$$

The symbol $*$ denotes a generic logical operator, which can be either a conjunction (`and`, $\wedge$) or a disjunction (`or`, $\vee$). The function $\Phi_k(x)$ is thus a logical formula, where predicates $P_{j,i} \in \mathcal{F}$ are applied to individual elements of $x$.

As nested lists $x_k \in \mathcal{X}$ may have different number of elements at each level of nesting, we need to specify how the action of $P \in \mathcal{F}$ depends on the length of $x_k = \langle x_{k,0}, \ldots, x_{k,N_k} \rangle$. We do so by introducing the operators $\gamma_i[x], i \in [0, \ldots, I]$:

$$\gamma_i[x_k] = \begin{cases} \Phi_{N_k}(x_{k,0}, \ldots, x_{k,N_k}), & \text{if } \gamma_i(x_k) \text{ is defined for length } N_k \\ 0, & \text{otherwise} \end{cases} \tag{4}$$

Based on the length of $x_k$, the operators $\gamma_i$ either produce some logical formula $\Phi_{N_k}$ (which in turn consists of some possibly recursive logical functions $P_{k,l} \in \mathcal{F}$) or `False`. We denote the set of all predicates that can be produced by $\gamma_i$ as $F_i$, which is also called a generating family. Note that $F_i(x_k)$ is also (a possibly recursive) logical function. Given $x_j \in \mathcal{X}$

$$F_i(x_k) = \gamma_i[x_k] = \begin{cases} \Phi_{N_k}(x_{k,0}, \ldots, x_{k,N_k}) = Q_{N_k,0}(x_0) * \cdots * Q_{N_k,N_k}(x_{N_k}) \\ 0 \end{cases} \tag{5}$$

The generating family $F_i(x_k)$ in Eq. 5 is a recursive function if $Q_{N_k,l}$ may take a value of $F_i(x_{k,l})$, for example.

We are now ready to formulate the PAG theorem. Given a set of polynomially computable non-recursive boolean functions $\mathcal{P} = \{P_0, \ldots, P_J\}$, given a set of boolean functions $\mathcal{F} = \{F_0, \ldots, F_I\}$ with the structure defined by Eq. 5, where

$$Q_{N_k,l}(x) = F_{k,l}(x) \in \mathcal{F} \quad \text{or} \quad P_{k,l}(x) \in \mathcal{P} \quad \text{or} \quad (x_l == e_l) \tag{6}$$

and each $\Phi_{N_k}(x_0, \ldots, x_{N_k})$ may contain no more than one copy of $F_k(x_j), \forall k \in [0, \ldots, I]$ for each $x_j, j \in [0, \ldots, N_k]$ in its definition. If $x \in \mathcal{X}$ is a list with finite depth and width, then the equation

$$F_i(x) == 1 \tag{7}$$

can be decided in a polynomial number of steps for any $F_i \in \mathcal{F}$. We will use this result to build a test system for large language models.

## 4 METHODS

Based on the results of the PAG theorem, we build a generator of problem instances with varying complexity. Each problem has the form of a set of generating families $\mathcal{F}$, a test object $x \in \mathcal{X}$ and the equality $F_i(x) == 1, F_i \in \mathcal{F}$. We call the set of generating functions a *condition* and the test object $x$ the *probe*. In the following sections, we will show how to express the *condition* as a Python program and present a probe generator algorithm, which allows the building of probes of any depth and any desired result when substituted in Eq. 7. The resulting Python code can be directly supplied to LLMs, and the response can be evaluated without human intervention, thus providing an automatic benchmark system.

## 4.1 CONDITION GENERATOR

The setup of the PAG theorem has to be mapped to a Python program to make a condition generator. We start by expressing logical formulas $\Phi$ in Eq. 3. Consider an input list with $N = 3$ elements $x = \langle x_0, x_1, x_2 \rangle$ of size 3. We first need to specify element-wise functions $Q_i, i \in [0, 1, 2]$. According to the requirements of the PAG-theorem, they can be either constant comparisons, non-recursive predicates $\mathcal{P}$ or possibly recursive generating families $\mathcal{F}$ (see Eq. 6). For simplicity, we do not use non-recursive predicates $\mathcal{P}$ here (an example of such predicate would be a Python function `bool()`, which returns `True` for any non-zero input or some other non-recursive function). We define $I = 2$ generating families called `is_member_0()` and `is_member_1()`. An example of the function $\Phi_N$ for $N = 3$ is shown below:

```
1  res = is_member_0(x[0]) and (x[1] == 2) and is_member_1(x[2])
```

Listing 1: $\Phi_3$ function example

To define a generating family, one must map inputs $x$ to different $\Phi$ functions based on the input length. An example of a set of two generative families called `is_member_0()` and `is_member_1()` is shown below. Notice that these functions may be mutually recursive.

```
1  def is_member_0(x):
2      if len(x) == 2:
3          res = (x[0] != 15) and (x[1] != 61)
4      elif len(x) == 3:
5          res = (is_member_1(x[0])) and (is_member_0(x[1])) and (x[2] ==
           49)
6      else:
7          return False
8      return res
```

```
1  def is_member_1(x):
2      if len(x) == 2:
3          res = (is_member_1(x[0])) and (is_member_0(x[1]))
4      elif len(x) == 3:
5          res = (x[0] != 46) and (x[1] == 95) and (x[2] != 7)
6      else:
7          return False
8      return res
```

Listing 2: An example set of generating families

We made several design choices to translate the requirements of the PAG theorem into concrete Python functions. Our code generator produces a set of $I$ recursive functions as shown in Lst. 2. Each function has if-blocks, which map inputs starting from length $N_i = 2$ to length $N_i^{max}$ to a corresponding logical formula $Q_{N_i}$. We use only `and` operators in logical formulas. Every logical formula can contain up to $b_i, b_i \leq N_i$ generating families $F_j \in \mathcal{F}$ in its definition, which is selected at random; the rest of the entries in $Q_{N_i}$ are defined as argument comparisons to random integer constants. Finally, each generating family $F_i$ has a logical formula of length $N_i^T$, which does not contain any calls to generating formulas $F_j \in \mathcal{F}$, e.g., completely composed of comparisons of the arguments to constants. In the example in Lst. 2 $N_0^T = 2$ for $F_0 = $ `is_member_0()` and accordingly $N_1^T = 3$ for $F_1 = $ `is_member_1()`. We found that having a terminal block in every $F_i \in \mathcal{F}$ is necessary for solving solutions of different nesting depths in Eq. 7.

We implemented the condition generator as described above using the template library Jinja2 (Ronacher, 2008). The generator produces random Python functions acting on lists $x$. The second half of the benchmark system is a probe generator, which is described in the next section.

## 4.2 PROBE GENERATOR

In this section, we devise an algorithm to build nested lists $x \in X$ satisfying Eq. 7. Notice that there are usually multiple solutions. For example, both `x0` and `x1` in Lst. 3

```
1      x0 = [-39, 21]
```

```
2    x1 = [[137, 95, -7], [93, 97], 49]
```

Listing 3: Solution examples

return `True` when passed to `is_member_0()` from Lst. 2, but the evaluation will take a different number of steps due to the different depth of the lists. To have an efficient benchmark, one would need to generate $x \in \mathcal{X}$ with a depth of nesting $d$, which will yield a desired result in Eq. 7. Such $x$ can be efficiently built.

To describe the algorithm, let us first consider a logical formula $Q^T_{N_i}$ completely made for comparing its arguments to constants. We call this formula a *terminal* formula.

$$Q^T_{N_i} = (x_0 == e_0) \wedge \cdots \wedge (x_{N_i} == e_{N_i}) \tag{8}$$

It is easy to find the argument which satisfies $Q^T_{N_i}$:

$$x = \langle e_0, \ldots, e_{N_i} \rangle \tag{9}$$

By changing a single entry in Eq. 9, one can build a non-solution. In general, *terminal* formulas can be composed of non-recursive predicates $P_j \in \mathcal{P}$ provided it is possible to solve $P_j(x) = 1$.

If a generating family $F_i$ produces $Q^T_{N_i}$, then $x$ from Eq. 9 is also a solution to $F_i(x) = 1$. Now consider a generating family $F_j$, which produces a logical formula $Q_{N_k}$ containing $F_i$:

$$Q_{N_k} = F_i(x_m) \wedge \widetilde{Q}_{N_k-1}(x_0, \ldots, x_{m-1}, x_{m+1}, \ldots, x_{N_k}) \tag{10}$$

here $\widetilde{Q}_{N_k-1}$ is a subformula of $Q_{N_k}$ without the term $F_i(x_m)$. If $\tilde{x} = \langle \tilde{e}_0, \ldots, \tilde{e}_{m-1}, \tilde{e}_{m+1}, \ldots, e_{\tilde{N}_k} \rangle$ satisfies $\widetilde{Q}_{N_k-1}$, then

$$y = \langle \tilde{e}_0, \ldots, \tilde{e}_{m-1}, \langle e_0, \ldots, e_{N_i} \rangle, \tilde{e}_{m+1}, \ldots, e_{\tilde{N}_k} \rangle \tag{11}$$

is a solution of depth one of $F_j(x) = 1$. In general, the solution of depth $d$ can be constructed with an algorithm outlined in Alg. 1.

---

**Algorithm 1** Probe generator

1: **function** SOLVETERMINAL($Q^T, r$)                                      ▷ Solves $Q^T(x) = r$
2:     $x \leftarrow \langle e_0, \ldots, e_N \rangle$, s.t. $Q^T(x) = r$
    **return** $x$
3: **end function**

4: **function** GENERATESOLUTION($\mathcal{F}, F_i \in \mathcal{F}, d, r$)    ▷ Finds $x$ of nesting level $d$, s.t. $F_i(x) = r$
5:     **if** $d = 0$ **then**                                  ▷ $x$ is a shallow list, solve terminal formula $Q^T$
6:         Find $Q^T \in F_i$
7:         $x \leftarrow$ SOLVETERMINAL($Q^T, r$)
8:         **return** $x$
9:     **else**                          ▷ $x$ is a nested list, recurse into a random non-terminal formula $Q_k$
10:         Find a random $Q_k \in F_i$, s.t. $Q_k = F_{j_0} \wedge \cdots \wedge F_{j_b} \wedge \widetilde{Q}_k$
11:         $y \leftarrow$ SOLVETERMINAL($\widetilde{Q}_k, 1$)                ▷ Solve a non-recursive part of the formula
12:         **for** $j \in [j_0, \ldots, j_b]$ **do**                          ▷ Breadth-first search-like iteration
13:             $y_j \leftarrow$ GENERATESOLUTION($\mathcal{F}, F_j, d-1, r$)
14:         **end for**
15:         $x \leftarrow \langle y_{j_0}, \ldots, y_{j_b}, y \rangle$
16:         **return** $x$
17:     **end if**
18: **end function**

---

The algorithm 1 is a breadth-first search-like recursive algorithm. Recall that every $F_i \in \mathcal{F}$ has a terminal formula $Q^T_i$ by our design choice a; hence we can always generate a shallow solution $x = \langle e_0, \ldots, e_N \rangle$ such that $F_i(x) = 1$. If we have to generate a nested solution of depth $d$, then we choose a non-terminal formula $Q_k \in F_i, Q_k = F_{j_0} \wedge \cdots \wedge F_{j_b} \wedge \widetilde{Q}_k$ at random, solve a non-recursive part $\widetilde{Q}_k$ of $Q_k$, and recurse into corresponding generating families $F_j, j \in [j_0, \ldots, j_b]$ until a required depth $d$ is reached. Notice that when generating non-solutions $\bar{x}, F_i(\bar{x}) = 0$, we choose not to satisfy only the terminal formulas at depth $d$. The latter guarantees that in order to verify $F_i(\bar{x}) == 1$, the evaluator will take at least $d$ steps to check the nested elements of $\bar{x}$.

## 4.3 EVALUATION

Having introduced the algorithms for data generation, we need to specify how they will be supplied to language models. In this work, we considered two kinds of models: conversational LMs and the models trained specifically for code completion. For conversational models, we used the following prompt:

```
1    What is the result of the following Python code?
2    ```python
3    {condition}
4    x = {probe}
5    print(is_member_0(x))
6    ```
7    Answer only `True` or `False`
```

Listing 4: Prompt for conversational LMs

In Lst. 4 the placeholder *condition* takes the definitions of the generative families $\mathcal{F}$, and *probe* takes the value of $x$. For code-specific models, there is no clear way to ask the model to evaluate some expressions. Instead, we relied on the ability of these models to enforce the correctness of Python expressions:

```
1  {condition}
2  x = {probe}
3  assert is_member_0(x) ==
```

Listing 5: Prompt for code completion LMs

The response of the language models is easy to verify automatically using the proposed prompts.

## 5 RESULTS

### 5.1 SETUP

We selected several state-of-the-art general purpose and code completion LMs for the experimental evaluation listed in Tab. 1. In the experiments, we compared the accuracy of LMs concerning each

Table 1: Tested models along with their versions

| Conversational LMs | Code completion LMs |
|---|---|
| GPT-3.5 [turbo-0125] (Ouyang et al., 2022) | PolyCoder [2.7B] (Xu et al., 2022) |
| GPT-4-turbo [2024-04-09] (Achiam et al., 2023) | CodeLlama [7B-Python-hf] (Roziere et al., 2023) |
| GPT-4o [2024-05-13] (gpt) | Starcoder-2 [3B] (Lozhkov et al., 2024) |
| Llama-3 [8B] (Dubey et al., 2024) | Stable Code [3B] (Pinnaparaju et al., 2024) |

parameter of the condition and probe generators. The combinations of parameters we used are listed in Tab. 2.

Table 2: Parameters of the example generator used in experiments. $I$ - total number of functions, $N_i^{max}$ - number of if-blocks in each function, $b_i$ - maximal number of calls to recursive functions in each if-block (branching number), $N_i^T$ - size of the *terminal* if-block, $d$ - nesting depth of the test list $x$.

| $I$ | $N_i^{max}$ | $b_i$ | $N_i^T$ | $d$ | Varying parameters |
|---|---|---|---|---|---|
| 2 | 2 | 1 | 2 | 2 | $I, N_i^{max}, d$ |
| 11 | 2 | 1 | 2 | 2 | $b_i$ |
| 2 | 12 | 1 | 2 | 2 | $N_i^T$ |

We generated 160 random problem instances with $F_i(x) = 1$ (*positive* instances) and the same number of instances with $F_i(x) = 0$ (*negative* instances) for each parameter combination of the

generator. We found that with this sample size, the standard deviation of all metrics is within 10%. Each model was tested using the same data. Additionally, we used the Bernoulli distribution as a baseline.

We used four binary classification metrics to compare the behavior of language models in the experiments: true positive rate (sensitivity, TPR = true positives/total predicted positives), true negative rate (specificity, TNR = true negatives/total predicted negatives), balanced accuracy BA = $\frac{\text{TPR}+\text{TNR}}{2}$ and the Youden's J statistic $J = \text{TPR} + \text{TNR} - 1$. Youden's statistic estimates the probability of an informed decision.

## 5.2 MODEL COMPARISON

We observed poor performance of every tested LLM in the benchmark across a wide range of generator parameters. Even for the simplest problem instances Lst. 6, balanced accuracy never exceeded 75%, despite the relative simplicity of the underlying verification process. Note that as each answer is either `True` or `False`, a simple random guess already achieves 50% accuracy. The accuracy of most small-scale models rapidly approached 50% with the increase of the probe's depth, the number of functions, and the number of if-blocks, with low sensitivity to the length of the terminal block. We also found that the accuracy of stronger models (GPT-3.5 and Codellama) increases with the number of recursive branches in functions $F_i(x)$, presumable because it makes easier to verify the probe $x$ by not checking the result of most of the if-blocks. The accuracy's dependence on the benchmark parameters is provided in Fig. 1.

```python
What is the result of the following Python code?
```python
{def is_member_0(x):
    if len(x) == 2:
        res = (x[0] != -20) and (x[1] != 1)
    elif len(x) == 3:
        res = (is_member_1(x[0])) and (x[1] == 20) and (x[2] == 30)
    else:
        return False
    return res
def is_member_1(x):
    if len(x) == 2:
        res = (x[0] == 65) and (is_member_0(x[1]))
    elif len(x) == 3:
        res = (x[0] != 24) and (x[1] != -15) and (x[2] == 37)
    else:
        return False
    return res}
x = {[3, -44]}
print(is_member_0(x))
```
Answer only `True` or `False`
```

Listing 6: An example set of a simple problem

Surprisingly, we found that an advanced model GPT-4-turbo performed worse than random and worse than its simpler versions, see Fig. 1. The observation that GPT-4-turbo is strongly biased on benchmark problems motivated us to check the behavior of LMs on positive and negative instances separately (TPR and TNR metrics). We found that every model in the benchmark demonstrated a significantly biased behavior on "`True`" and "`False`" subsets of problems, with some models (PolyCoder, StarCoder) generating a constant answer regardless of the condition used. GPT-3.5 demonstrated the slightest bias of all tested LLMs, followed by LLama3 and GPT-4o. Again, GPT-4-turbo demonstrates a strange bias towards incorrect answers. An example of model behavior with varying probe depth on separate problem subsets is shown in Fig. 2, and additional examples can be found in Appendix B. In all experiments, Youden's $J$ statistic did not exceed 0.45, with typical values of around 0.2, suggesting that LLMs can barely use the condition part of the problem to predict the answer.

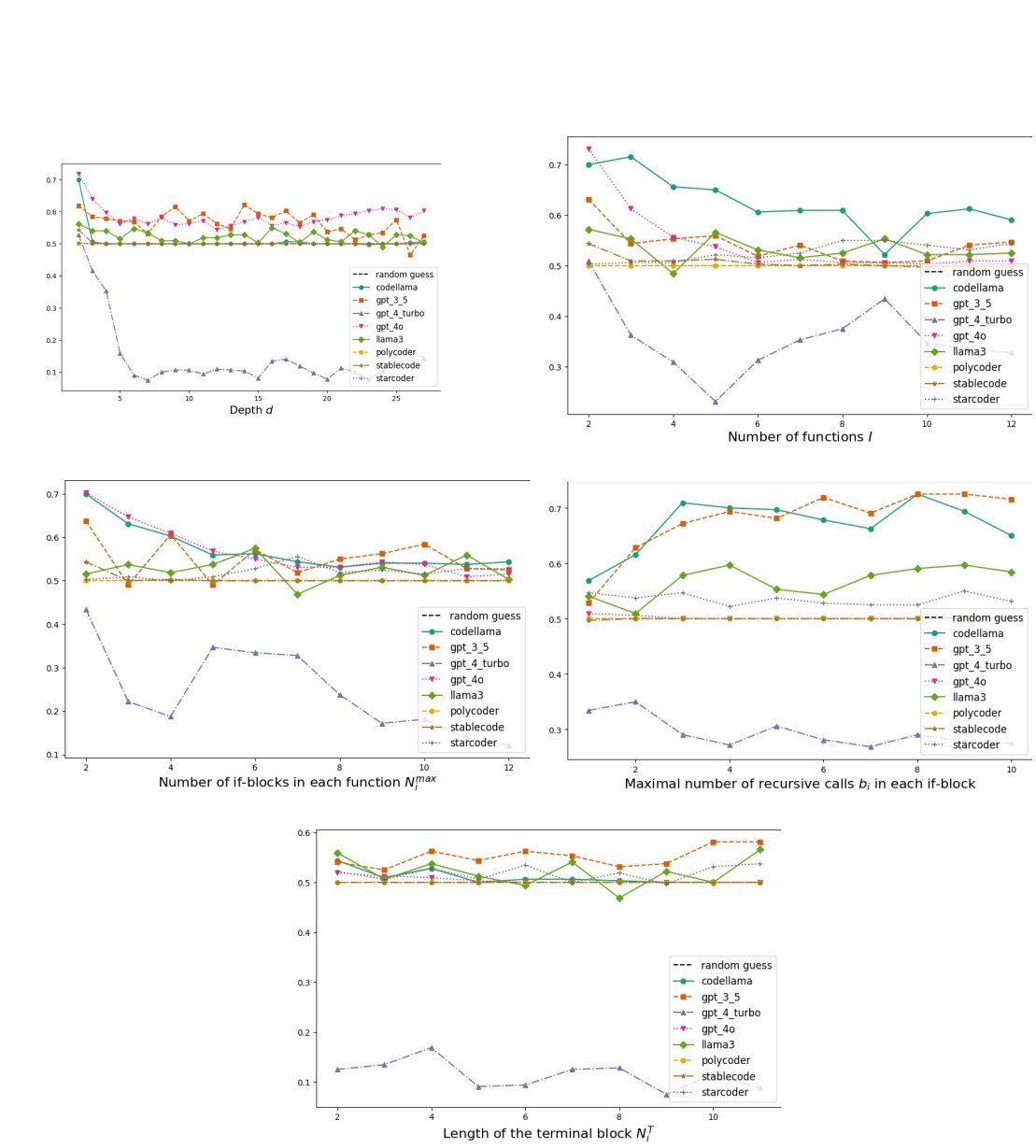

Figure 1: Sensitivity of LLM's balanced accuracy to different parameters of the benchmark

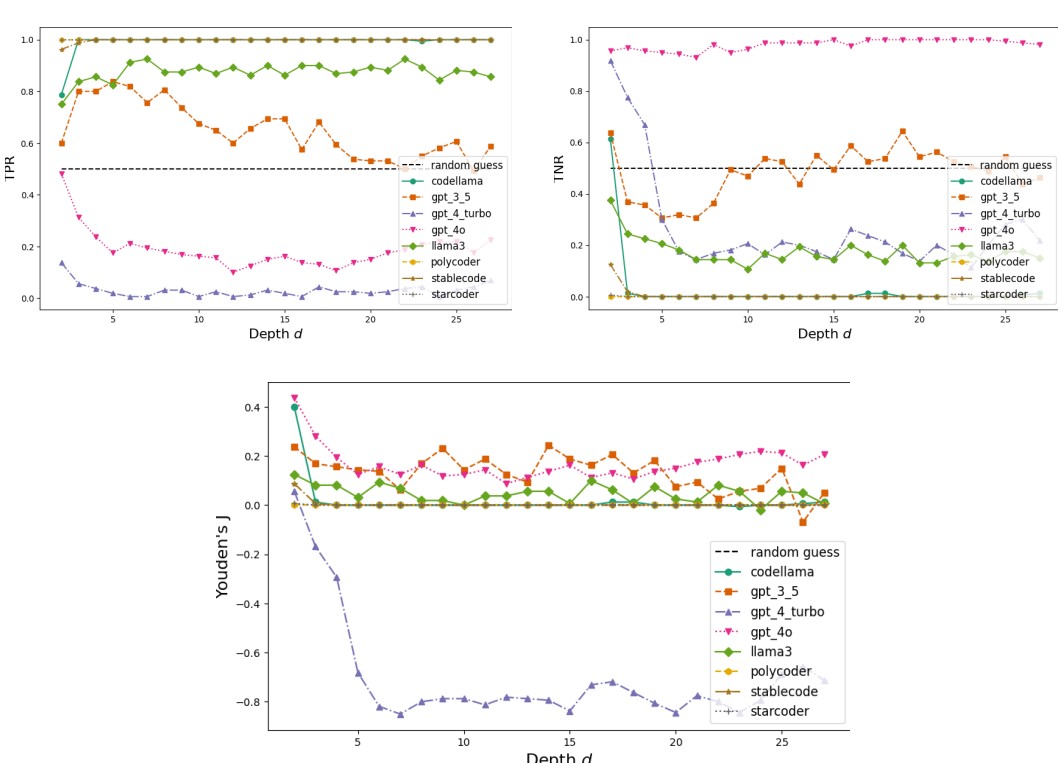

Figure 2: TPR, TNR and Youden's $J$ statistic dependence on the depth of probe $x$. Despite having almost perfect TPR, StarCoder, Polycoder, and StableCode have zero TNR, suggesting a highly biased behavior.

# 6 DISCUSSION

The construction of benchmark systems is essential steps in the evaluation of LLMs performance. Our reasoning benchmark is based on the results of the polynomial analog of Gandy's classical fixed point theorem and provides means for principled LLMs evaluation. We found that even smallest problem instances are challenging for state of the art LLMs.

The unexpected finding was a significant bias inherent to every tested LM. No model demonstrated an expected Bernoulli distribution-like behavior when tested on uniform positive or negative problem instances, with GPT-3.5 coming closer to the expected distribution. This observation suggests that LLMs may have difficulty with the generation of random data. As the most extreme case, biased behavior renders GPT-4-turbo the worst performer in our test, suggesting that bias increases with the model scale.

Contrary to what may be expected, models trained to understand code demonstrated worse performance than general-purpose LLMs. We plan to adapt our benchmark to natural language formulation to make it even more natural for general purpose LLMs.

Overall, our experiments demonstrate an unusual behavior of LLMs for recursive code-understanding tasks. None of the models can consistently solve even simple problem instances despite the existence of a deterministic algorithm with quadratic complexity (Depth-First search with backtracking). Our findings stipulate the revision of language model training procedures to eliminate biases we observed, particularly in contexts where high accuracy and reliability in reasoning are required.

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

# A  THE POLYNOMIAL ANALOGUE OF GANDY'S FIXED POINT THEOREM

The polynomial analog of Gandy's fixed point theorem (PAG-theorem) (Goncharov & Nechesov, 2021b; Nechesov, 2023) is based on the classical Gandy's fixed point theorem (Ershov, 1996), which is important not only in mathematical logic but also in programming. The classical Gandy's theorem states that if we have a fixed model:

$$\Omega = \langle \mathcal{N}, 0, +, \cdot, \leq \rangle \text{ of the signature } \sigma_0 = \langle 0, +^2, \cdot^2, \leq^2 \rangle$$

Denote by $\sigma^*$ the extension of the signature $\sigma_0$ obtained by adding symbols for all $\Sigma$-functions (Ershov, 1996) on $\Omega$ and constant symbols for all elements from $\mathcal{N}$, and let $\Omega^*$ denote the corresponding enrichment of $\Omega$.

Define the operator $\Gamma^{\Omega^*}_{\Phi[\overline{x}]}$ as next:

$$\Gamma^{\Omega^*}_{\Phi[\overline{x}]}(Q) = \{(e_1, \ldots, e_{k-1}) \mid \langle \Omega^*, Q \rangle \models \Phi(e_0, \ldots, e_{k-1})\} \cup Q \tag{12}$$

where $\models$ symbol means checking the truth of the formula $\Phi(e_0, \ldots, e_{k-1})$ with initialized variables on the model $\langle \Omega^*, Q \rangle$, and $Q \subseteq \mathcal{N}^k$ is a truth set of an extendable predicate, $(e_1, \ldots, e_{k-1}) \in Q$.

We associate the next sequence of the sets of the truth of predicates:

$$\Gamma_0, \Gamma_1, \ldots, \Gamma_\alpha \text{ where } \alpha \text{ is an ordinal.}$$

with the monotone operator $\Gamma^{\Omega^*}_{\Phi[\overline{x}]}$ as follows:

$$\Gamma_0 = \emptyset, \Gamma_{\alpha+1} = \Gamma^{\Omega^*}_{\Phi[\overline{x}]}(\Gamma_\alpha) \text{ for none limit ordinal}, \ldots, \Gamma_\alpha = \cup_{\beta < \alpha} \Gamma_\beta \text{ for limit ordinal}$$

Let $\alpha$ be the smallest ordinal such that $\Gamma_{\alpha+1} = \Gamma_\alpha$ then $\Gamma^* = \Gamma_\alpha$ is a smallest fixed point.

**Theorem 1** (Gandy's fixed point theorem)
Let $\Phi(P^+)$ be a $\Sigma$-formula of the signature $\sigma^* \cup \{P^{(k)}\}$ in which the predicate symbol $P$ enters

positively and $x_0, \ldots, x_{k-1}$ be a list of the different free variables of the formula $\Phi$ then the smallest fixed point $\Gamma^*$ of the operator $\Gamma^{\Omega^*}_{\Phi[\overline{x}]}$ is a $\Sigma$-predicate on $\Omega^*$.

In the PAG-theorem, several conditions from the classical Gandy's fixed point theorem were modified or strengthened so that the fixed point of the operator remains a polynomially computable (P-computable) predicate.

To prove this, a P-computable hereditary-finite list superstructure $HW(\mathfrak{M})$ of signature $\sigma$ was chosen as the basic model (Goncharov & Nechesov, 2021b). The base set $HW(M)$ contains hereditary-finite lists, which are inductively constructed from the basic elements of the base set $M$ of the model $\mathfrak{M}$ of the signature $\sigma_0 \subset \sigma$. The basic list operations for $HW(\mathfrak{M})$ are:

- $head(x)$ - operation of taking the last element of a list $x$

- $tail(x)$ - operation of removing the last element from a list $x$

- $cons(x, y)$ - operation of adding to the end of a list $x$ of a list $y$

- $conc(x, y)$ - concatenation operation of two lists $x$ and $y$ respectively

There are also relations:

- $x \in y$ - "$x$ is an element of $y$"

- $x \in y$ - "$x$ is an initial segment of $y$"

Formulas of the first-order logic are considered only with the bounded quantifiers $\forall x \in t$, $\exists x \in t$, $\forall x \subseteq t$, $\exists x \subseteq t$, where $t$ is a standard term.

In order to apply the PAG theorem generally, it is necessary to construct a P-computable GNF system based on the following components:

- A finite alphabet $\Sigma$.

- An extended alphabet $\Omega$.

- A special logical language L where L-formulas and L-programs are defined

- P-computable hereditary-finite list superstructure HW(M) of the signature $\sigma$.

- Finite sets of extendable predicates $P_1, \ldots, P_n$.

- Generating families of formulas $F_{P_1}, \ldots, F_{P_n}$.

- P-computable functions $\gamma_1, \ldots, \gamma_n$ that, given an element of HW(M), construct suitable generating formulas or return False.

- and other conditions from (Goncharov & Nechesov, 2021b)

**Theorem 2** (Polynomial analogue of Gandy's fixed point theorem)
Let $G$ be a p-computable GNF-system then the smallest fixed point $\Gamma^*$ of the operator $\Gamma^{HW(\mathfrak{M})}_{F_{P_1}, \ldots, F_{P_n}}$ is P-computable.

The results obtained in the PAG-theorem allow us to inductively define sets of objects of varying complexity using generating families of L-formulas and, at the same time, guarantee their recognizability in polynomial time. Inductivity upwards essentially defines recursivity downwards, i.e., the algorithm that checks whether an object belongs to a class of objects, working with various list encoded objects, each time splits the list into elements and recursively checks each element for consistency with one of the families of generating formulas. This entire process can be programmed so that the question of whether an element belongs to a set is solved in polynomial time (Goncharov & Nechesov, 2021a; 2022).

**Corollary 1** Let $G$ is a P-computable GNF-system. Let the computational complexity of all basic functions, predicates, and functions $\gamma_i(x)$ in $G$ be at most $O(|x|^p)$. Then the computational complexity of the smallest fixed point $\Gamma^*$ is at most $O(|x|^{p+2})$.

# B    ADDITIONAL DATA

Here, we provide additional benchmark data. Pairs of TPR and TNR plots for various benchmark parameters are listed in Fig. 3.

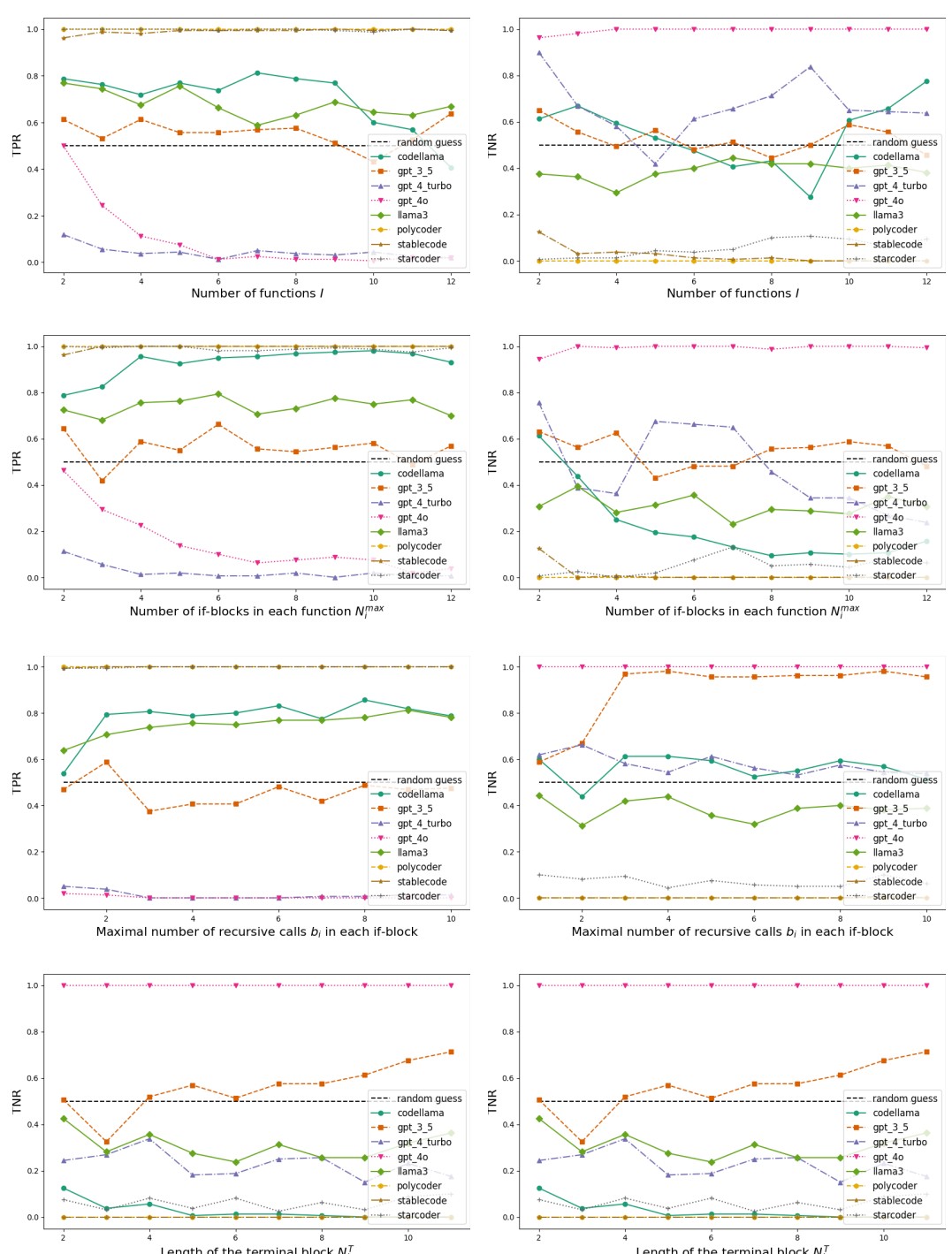

Figure 3: Pairs of TPR and TNR plots concerning different parameters of the benchmark

Youden's statistic behavior depending on the parameters of the benchmark is shown in Fig. 4

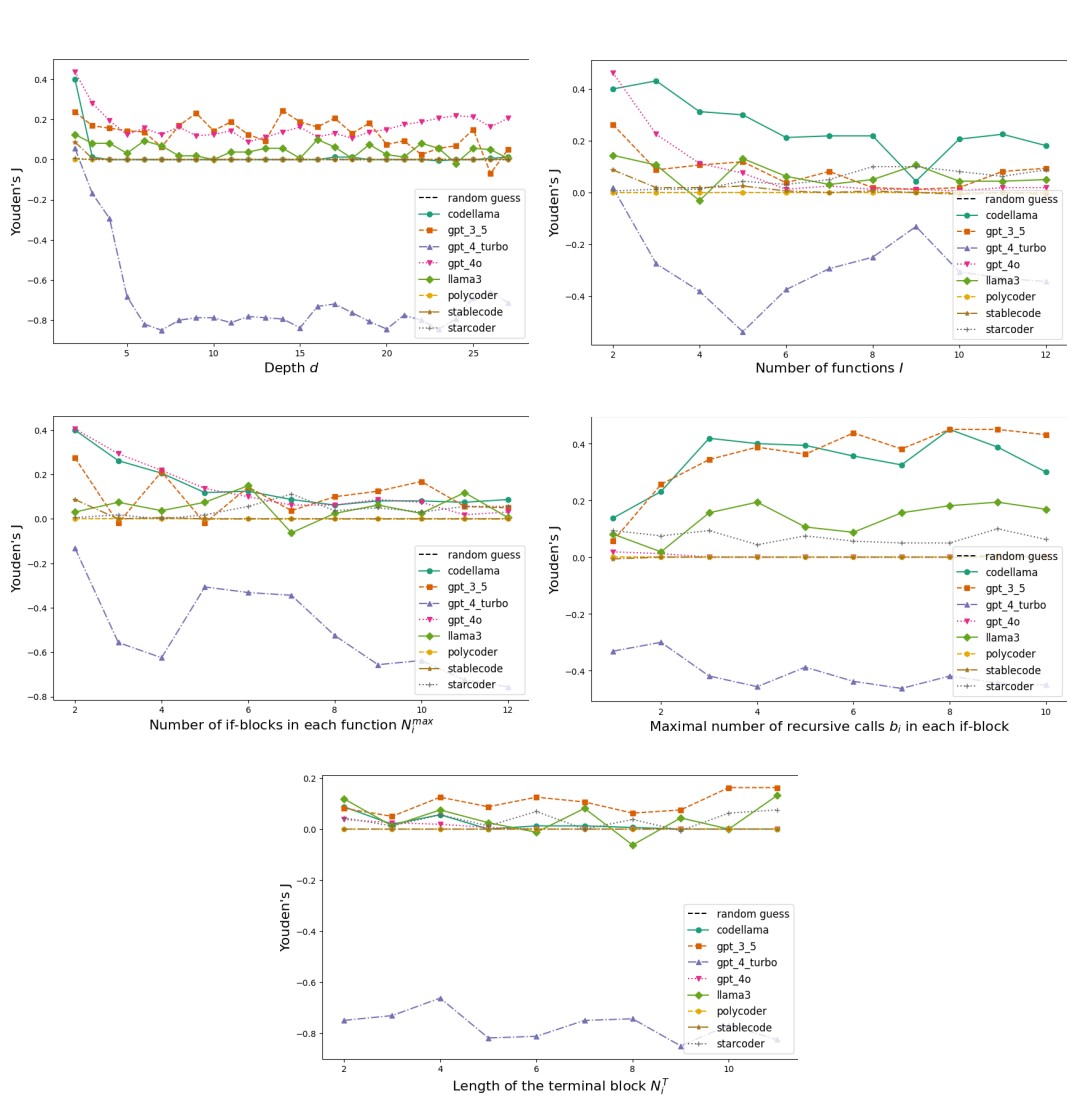

Figure 4: Youden's $J$ depending on the parameters of the benchmark

