# OpenReview forum: "Rethinking logic in AI: A novel benchmark inspired by polynomial analogue of Gandy's fixed point theorem"
_ICLR.cc/2025/Conference — Submitted to ICLR 2025_

### Official Review · Reviewer_ocgh · 2024-11-01

**Soundness:** 2
**Presentation:** 2
**Contribution:** 1
**Rating:** 3
**Confidence:** 3

**Summary:**

This paper introduces a new benchmark dataset for evaluating the reasoning ability of LLMs, based on the PAG theorem. The problems in the dataset contain a theorem that returns boolean output based on a given nested list. The dataset is created using a condition generator and a probe generator to generate the theorems and the nested list instances respectively. The theorem and the assignment are represented in text as Python functions and lists. The paper then prompts LLMs to predict whether a theorem is satisfied or not by the given nested list, and finds that state of the art LLMs are unable to when prompted in a zero shot setting.

**Strengths:**

The proposed dataset can be generated parametrically, which varies its difficulty level.

The proposed dataset is difficult for existing LLMs.

The dataset generators are well-defined.

**Weaknesses:**

To me, the dataset is not strongly motivated. I agree that new reasoning benchmark datasets are important for LLMs, but why this specific dataset is needed is unclear. What is the advantage of this dataset over, for example, a set of problems that include SAT formulas in CNF and the set of assignments to the variables, where the LLM is asked to verify if the solution is valid?

It seems that the LLMs are only evaluated on this dataset using zero-shot prompting. Extensive literature has shown that LLM reasoning abilities can be significantly enhanced by chain-of-thought [1], tool-using [2], self-consistency [3] etc. Therefore, it is hard to draw a definitive conclusion on LLM reasoning abilities from zero-shot prompting results alone.

By representing the problem in Python code, the dataset is limited to evaluating LLM’s reasoning ability in the specific context of executing Python, which may not represent LLM’s reasoning ability in the general case. For example, by asking the conversational LLM to output the result of print(is_member_0(x)), the LLM needs to understand the Python syntax on top of the logic of whether x satisfies f or not.

[1] Wei, Jason, et al. "Chain-of-thought prompting elicits reasoning in large language models."

[2] Schick, Timo, et al. "Toolformer: Language models can teach themselves to use tools."

[3] Wang, Xuezhi, et al. "Self-consistency improves chain of thought reasoning in language models."

**Questions:**

The problem is prompted to the LLMs in the format of python code. Why not represent the problems using text? Were other representations considered to represent the problems?

Were there any further analysis performed on the results shown in Figure 1? Would simply negating the solution from GPT-4-turbo achieve 90%+ accuracy? Does a model getting the wrong answer 90% of the time mean it is consistently doing (wrong) reasoning internally?

---

> ### Author Response · Authors · 2024-11-24
> **Dear reviewer, thank you for your critical comments. We really hope for your positive decision on the article after our responses.**
>
> Dear Reviewer, thank you for valuable suggestions. Here are the answers to your questions:
>
> “How is this benchmark motivated and what is the advantage over SAT formulas?”
>     The motivation of the benchmark is the automatic generation procedure, tunable complexity and a correlation of the LLM performance with other (non-generated) reasoning benchmarks. Following your suggestion of the SAT problem we found the works [1] https://arxiv.org/abs/2408.07215 [2] https://arxiv.org/abs/2312.14890 which use similar NP problems in their benchmark and cited them. Although these benchmark are very useful, our approach has the following advantages:
> a. Our approach allows for exact parametric control of the example complexity. For NP problems it is hard to generate problems of specified complexity, as, for example, two instances of NP problem of the same size can have a very different complexity. Additionally, it is not possible to generate very large NP problems with known and hard to find solutions. We have no limit on the problem size.
> b. Our approach generates recursive problems, which turn out difficult for LLMs. There is a limited number of works studying recursion in LLMs.
> c. We can test both general and code-generating LLMs on the same problems.
>
> In the work with the 3-SAT problem (https://arxiv.org/abs/2408.07215) using LLM occurred with a small number of free variables (no more than 10). That is, a 3-SAT problem in which it is possible to find a solution manually by enumeration. It turned out that even with such a small number of parameters, LLMs cope poorly with them. As expected, with an increase in parameters, their capabilities sharply decrease. At the same time, the 3-SAT problem is NP-complete, which raises the question: why study it if we have not yet dealt with the class P, which is much weaker and where there are still a lot of problems even with our recursive benchmark, the complexity of which is always P.
>
> “LLM reasoning abilities can be significantly enhanced by chain-of-thought…”
>     Dear reviewer, we agree with you that the right train of thought can help llm solve the problem better. But the problem is, which train of thought to choose? Because the number of possible paths grows exponentially. And is it really necessary to help llm solve problems from class P? After all, we already feed the inductive algorithm in Python as input, which they can use as a solution.
>     In the next paper, within the framework of your comment, we will do a deep analysis of this direction. This is especially important for the classes NP-complete and NP-hard, but here our tool is a polynomial analogue of Gandy's theorem, which allows us to create recursively defined sets of objects and at the same time not to jump out of the class of polynomiality P.
> Note that chain-of-thought is not available for code LLMs. Additionally, the tool use (an interpreter tool specifically) was shown to significantly improve reasoning in LLMs in multiple works, but spoils the benchmark. Our problems can be solved by a single call of the interpreter. As for the self-consistency, we average every answer over 10 LLM calls for each data point.
>
> “The problems are in the format of Python code. Why not represent the problems using text?”
>       We tried different problem formats in the beginning, but switched to code to test both general and code-generating LLMs and to simplify response parsing. We save the study of different representations for future work. Please note that it may be quite difficult to formulate this benchmark in easy to read natural language.
> “Would negating the solution from GPT-4-turbo achieve 90%+ accuracy?”
>      It turns out that some LLMs have a specific bias towards a constant answer, e.g. their answer does not depend on the condition part of the problem.
> "Does a model getting the wrong answer 90% of the time mean it is consistently doing (wrong) reasoning internally?"
>    These tests are primarily aimed at identifying thinking abilities based on solving recursive problems, if LLM solves the problem correctly, then yes, LLM can think in our understanding. Otherwise, it is not clear.
>
> If we've resolved your concerns, we kindly ask that you increase the score. Otherwise, we would like to continue the discussion.
>
>
> PS: Below we would like to describe the importance of this benchmark to you:
>
> In this paper, we propose to focus on a unique combination of problems that lie in the class P and at the same time admit a recursive description. If LLMs cannot solve these problems, then most often it is pointless to move on to more complex problems from the NP-complete and NP-hard classes, since their complexity becomes higher and at the same time most of them also admit a recursive description. In this investigation, we focus on the PAG theorem, demonstrating that even within this framework, LLMs encounter significant difficulties.

---

> > ### Author Response · Authors · 2024-11-25
> > **Representability of the HornSAT problem using a modified Polynomial Analogue of Gandy's theorem.**
> >
> > We would also like to share with you the information that we have managed to describe the HornSAT problem, which lies in the class P, using a modified polynomial analogue of Gandy's theorem. Thus, we have shown that HornSAT is a special case for those problems that can be described using our theorem.
> >
> > We had to transfer the main concepts of the PAG theorem (https://doi.org/10.3390/math9172102) from nested lists to the set of horn formulas, and also had to abandon generating families in favor of the set of horn formulas. This was achieved by redefining the gamma function, which now takes some horn formula as input and produces another shorter horn formula according to a recursive PTIME-algorithm for Horn satisfiability: (https://en.wikipedia.org/wiki/Horn-satisfiability).
> >
> > We have obtained a new modified polynomial analogue of Gandy's theorem and constructed a monotone operator G with the fixed point property. The operator G acts on the set of all subsets of horn formulas in the set of all subsets of horn formulas. In this case, the smallest fixed point Γ of the operator G is PTIME-computable. Without loss of generality, we can assume that the HornSAT problem is a special case of a modified polynomial analogue of Gandy's theorem.

---

> > > ### Comment · Reviewer_ocgh · 2024-11-26
> > >
> > > Thank you for the detailed response. I now have a better understanding of the paper’s motivations.
> > >
> > > However, I am still not convinced that the paper has significant enough contribution. The main reason behind my recommendation is that, the paper aims to examine LLM’s logical reasoning abilities, but does not evaluate them using the state-of-the-art prompting approaches that have been shown to improve reasoning performance significantly. Therefore, I don’t think the conclusions drawn by the paper, such as “current state-of-the-art LLMs fail to reliably solve even the most straightforward cases”, are well justified.

---

> > > > ### Author Response · Authors · 2024-11-27
> > > >
> > > > Dear reviewer, regarding the fact that good promts can enhance the reasoning abilities of large language models, how can this be proven? The fact that we are adding the phrase "think step by step" does not provide any guarantees that LLM will start solving better. We believe that systems like ChatGPT, in which thousands of man-hours and billions of dollars have been invested, should be able to solve these problems without improved promts, and there should be some mathematical justification for this, and not just tests. But we listened to your comment and will try to explore the state-of-the-art prompting approaches  in the future.
> > > >
> > > > But in our work, promts are the algorithm for solving the problem in Python, which we feed to the LLM input in order to specify an inductively defined set of objects. And then, we check the inclusion of some object in this set.
> > > >
> > > > Moreover, when building trusted artificial intelligence, we must understand that LLM does not make mistakes on the simplest problems (problems from class P). And there will be no improved promts there.
> > > >
> > > > "Therefore, I don’t think the conclusions drawn by the paper, such as “current state-of-the-art LLMs fail to reliably solve even the most straightforward cases”, are well justified."
> > > >
> > > > Taking into account your comments, we will try to soften our conclusions about the cognitive abilities of LLMs!
> > > >
> > > > An important point that we would like to note is that our benchmark uses the PAG theorem, which allows us to analyze not just one specific problem from the class P like 2-SAT, HornSAT, etc. Using the PAG theorem, we can analyze the entire class of problems P as a whole, both recursive and non-recursive (any problem in the class P can be made recursive by simulating the work of a Turing machine implementing it on the first polynomial steps).
> > > >
> > > > We believe that the uniqueness of our benchmark is in its universality, which allows us to work in the class P on problems of varying complexity.
> > > >
> > > > We express our hope that you will reconsider your decision, and our paper will be accepted for publication at the conference. Thank you for your attention to this matter!!!

---

> > > > > ### Author Response · Authors · 2024-11-28
> > > > > **The main contribution of the benchmark is its universality in the class P, since we can describe both the 2-SAT problem and the p-complete HornSAT problem with it. Thus, we can describe any problem in the class P, using the PAG theorem.**
> > > > >
> > > > > Dear reviewer, we understand that the terms of our cooperation are coming to an end. Therefore, we would like to summarize our benchmark. The main contribution of the benchmark is its universality in the class P, since we can describe both the 2-SAT problem and the p-complete HornSAT problem with it. Thus, we can describe any problem in the class P, using the PAG theorem. Thus, we can make a serious leap in the development of LLM. We really hope that you will appreciate the mathematical aspects that are laid down in the article and change your decision in a positive direction and we will see you again at the ICLR 2025 conference!

---

> ### Author Response · Authors · 2024-11-30
> **What do you think about our contribution to the construction of a p-complete benchmark using the PAG theorem for LLM?**
>
> Dear Reviewer, since there is still time and we have described how a polynomially complete benchmark in class P for LLM is constructed mathematically using Polynomial Analogue of the Gandy's fixed point theorem, what do you think about our contribution to this direction? Is it really insignificant?

---

### Official Review · Reviewer_xkwH · 2024-11-03

**Soundness:** 3
**Presentation:** 2
**Contribution:** 2
**Rating:** 6
**Confidence:** 2

**Summary:**

This paper introduces a novel benchmark for evaluating the logical reasoning capabilities of Large Language Models (LLMs).

The proposed approach involves generating [object, condition] pairs, where objects are nested lists of elements (for example, numbers) and conditions are expressed with recursive functions. The task of the LLM is to determine whether the object satisfies all conditions.
The source code of the benchmark is publicly available.

The main finding of the paper is that current state-of-the-art LLMs fail to reliably solve even the most straightforward cases, thus pointing to significant limitation in their logical reasoning capabilities.

======================================================

Update after rebuttal: in the light of the discussion during the rebuttal period, I am happy to raise my score.

**Strengths:**

1) The challenge of assessing precisely the reasoning capabilities of LLMs is extremely important to the development of LLMs themselves. This paper makes a contribution in this sense by introducing what looks like an interesting benchmark.
However, why the proposed benchmark is actually important is not developed enough in the submission.

2) The experimental evaluation is rather in-depth, with several important LLMs considered, both proprietary and publicly available.

**Weaknesses:**

1) The paper is only 8 pages long, two pages shorter than the page limit.
It is not clear why the authors didn't include some of the material appearing in the appendix, for instance.

2) The authors define their class of problems through the Polynomial Analogue of Gandy’s Fixed Point Theorem (which I am not familiar with). However, it is not clear why this specific class is so important as to represent a benchmark for LLMs.

Also, the authors might consider to clarify the relation with standard propositional Boolean logic (which I assume is the non-nested version).

3) The related literature is not discussed at any length. It almost looks like there has not been any other effort to assess the reasoning capabilities of LLMs, including logic-based reasoning.

**Questions:**

1) Why is the class of problems considered in this submission so relevant as to represent a benchmark for LLMs?

2) How does this contribution compare to the current state of the art?

---

> ### Author Response · Authors · 2024-11-24
> **Dear reviewer, thank you for the enormous work you have done. We hope our answers will help you reconsider the value of the article.**
>
> Dear Reviewer, thank you so much for an extensive analysis of the paper! Here are the answers to your questions:
> 1) “The paper is only 8 pages long..”
>     We believe that any complexity should be justified and try to avoid overwhelming the reader’s attention span. As you mentioned, the full size of materials is 14 pages, some extra space was used to address questions of the Reviewers. We increased the volume of the article to 10 pages without the list of references and appendix, the total number of pages is 16.
> 2) “It is not clear why this class of problems is important..”.
>     We believe that our benchmark has four important advantages:
>     a. It allows effective generation and verification of problems of any size and complexity, thus ensuring applicability to future LLMs.
>     b. The randomness of the problems should protect against memorization of particular problem instances.
>     c. Due to the ability to tune example complexity quantitatively, it allows to study the regularity and asymptotic behavior of LLM’s accuracy.
>    d. All problems are recursively definable and their complexity does not leave the class P. This can be achieved due to the PAG theorem
> 3) “The related literature is not discussed at any length...”
>     We struggled with finding similar works. However, we added a “Related works” section to overview other benchmarking systems.
>    The closest articles to ours is
>     [1] https://arxiv.org/abs/2312.14890
>     [2] https://arxiv.org/abs/2408.07215
>     [3] https://arxiv.org/abs/2406.02061
>     which we found following another Reviewer’s suggestion. Compared to this work our algorithm
>     a. provides exact control over the complexity of the problem instances
>     b. Consists of recursive problems, which are found hard for LLMs
> 4) “Why is this class of problems important to represent a benchmark?”
>     Logical reasoning is a critical aspect of LLM analysis, and in this article we revisit the logic behind LLM operations using a benchmark that is based on the findings of the polynomial analogue of Gandy’s fixed point theorem. Recursion, a concept that has been relatively underexplored, lies at the heart of logical reasoning, making it an essential topic for discussion. We believe that our choice of title is appropriate, but we welcome critical feedback on this matter.
> 5) “How does this contribution compare to the current state of the art?”
>     In this paper, we propose to focus on a unique combination of problems that lie in the class P and at the same time admit a recursive description. If LLMs cannot solve these problems, then most often it is pointless to move on to more complex problems from the NP-complete and NP-hard classes, since their complexity becomes higher and at the same time most of them also admit a recursive description. To navigate through recursive structures while remaining within the confines of class P, the PAG theorem (https://doi.org/10.3390/math9172102) and FPAG theorem (https://doi.org/10.3390/math12213429) prove invaluable, enabling us to define sets and functions that can be described recursively with polynomial complexity. In this investigation, we focus on the PAG theorem, demonstrating that even within this framework, LLMs encounter significant difficulties in determining whether an object belongs to an inductively defined collection of entities. The uniqueness of the benchmark is achieved due to its unique properties - in essence, we are trying to solve recursively defined problems, but at the same time do not jump out of the class P.
>
> If we've resolved your concerns, we kindly ask that you increase the score at least to the level of passing the article submission threshold. Otherwise, we would like to continue the discussion.

---

> > ### Comment · Reviewer_xkwH · 2024-11-26
> >
> > Thanks to the authors for their rebuttal.
> >
> > However, I still have doubts about the significance of the contribution.
> > Also, in the rebuttal to my question about the related work, the authors say "[w]e struggled with finding similar works." But benchmarks for LLMs is actually a very active area of research, including benchmarks for symbolic/logical/mathematical reasoning.
> > So, it is not entirely clear to me what the authors mean when they say "similar".

---

> > > ### Author Response · Authors · 2024-11-27
> > > **The uniqueness of our benchmark in LLM training and related works**
> > >
> > > Dear Reviewer, an important point that we would like to note is that our benchmark uses the PAG theorem, which allows us to analyze not just one specific problem from the class P like 2-SAT, HornSAT, etc. Using the PAG theorem, we can analyze the entire class of problems P as a whole, both recursive and non-recursive problems (any problem in the class P can be made recursive by simulating the work of a Turing machine implementing it in the first polynomial steps).
> > >
> > > Due to the fact that HornSAT is a P-complete problem, and the HornSAT problem can be described using a Polynomial Analogue of the Gandy's fixed point theorem, our benchmark covers the entire class P. This allows us to study the cognitive abilities of LLMs both in solving the simplest and the most complex problems in P.
> > >
> > > As for related works, indeed, there are very few works on benchmarks that rely on some serious mathematical theorems. Moreover, we did not find any works that would cover exactly one complexity class, as we do when using PAG theorem. But during our discussion, we realized that benchmarks for solving such problems as HornSAT and 2-SAT can be considered really close to our task, but only as special cases. Therefore, in the article, we reflected similar works in the section Related work.
> > >
> > > We believe that this is what makes our benchmark unique, and it can be a serious contribution to understanding how LLMs solve problems from the P class. If further results will be positive, then we can move on to problems from the NP class.
> > >
> > > We express our hope that you will reconsider your decision, and our paper will be accepted for publication at the conference. Thank you for your attention to this matter!!!

---

> > > > ### Author Response · Authors · 2024-11-28
> > > > **The main contribution of the benchmark is its universality in the class P, since we can describe both the 2-SAT problem and the p-complete HornSAT problem with it. Thus, we can describe any problem in the class P, using the PAG theorem.**
> > > >
> > > > Dear reviewer, we understand that the terms of our cooperation are coming to an end. Therefore, we would like to summarize our benchmark. The main contribution of the benchmark is its universality in the class P, since we can describe both the 2-SAT problem and the p-complete HornSAT problem with it. Thus, we can describe any problem in the class P, using the PAG theorem. Thus, we can make a serious leap in the development of LLM. We really hope that you will appreciate the mathematical aspects that are laid down in the article and change your decision in a positive direction and we will see you again at the ICLR 2025 conference!

---

### Official Review · Reviewer_SDXQ · 2024-11-03

**Soundness:** 3
**Presentation:** 3
**Contribution:** 2
**Rating:** 5
**Confidence:** 4

**Summary:**

The authors study the reasoning capabilities of LLMs by studying their capabilities to decide recursively defined properties of nested lists. They formulate the properties to be tested so that they are polynomial time testable. They implement a generator that generates these problem instances with varying complexity; pairs consisting of a nested list and recursively defined property. They consider two kinds of encodings of their problems to LLMs; inputs to conversational LMs and inputs to code completion LMs. They then compare the reasoning capabilities of different LLMs.

They found out that the LLMs they tested were not capable of deciding the recursive properties.

**Strengths:**

It is interesting to study the capabilities of LLMs using mathematically defined objects.  It is a nice idea to study whether LLMs can decide PTIME-computable recursive problems before tackling more complex ones.

**Weaknesses:**

In summary, the authors study the capabilities of LLMs by studying how well they can decide recursively defined properties of nested lists. While to have understanding on the capabilities of LLMs is important, the paper itself seems to me more like a good student project to showcase the limits of the current capabilities of LLMs, than a research paper ready to be published. In particular, the main contribution of the paper is the idea to study capabilities of LLMs by inputing pairs consisting of nested lists and recursively defined properties of those lists, the task being to decide whether the list satisfied the properties given. The authors then test various LLMs on the inputs they generate.

The theoretical contribution of the submission is to consider recursively defined properties and inputs to decide, and the technical contribution is to generate these inputs and tabulate the results with respect to different LLMs. I do not think that either contribution of the paper suffices for a publication in a top general conference in machine learning.

**Questions:**

None.

---

> ### Author Response · Authors · 2024-11-24
> **Dear Reviewer, Thank you for your critical comments. We hope our answers will increase the rating of our article.**
>
> Dear reviewer, these are not just recursively defined sets of nested lists. There is serious mathematics behind all of this, such as the PAG theorem (Polynomial Analogue of Gandy’s Fixed Point Theorem https://doi.org/10.3390/math9172102), which allows us to define recursively definable sets without going beyond polynomial complexity, i.e. to remain in the P class. Since most other recursive descriptions for problems like 3-SAT or the decision version of the TSP have NP-complete complexity, our P class is much narrower and it makes sense to study it before moving on to more complex classes NP-complete, NP-hard, and etc. Most other benchmarks try to cover all complexity classes (P, NP-complete, NP-hard) (https://arxiv.org/abs/2312.14890) , which in our opinion is questionable. Since the P class has enough recursive problems that LLMs cannot solve.
>
> If you are satisfied with our answers, we will be very grateful if you reconsider your decision to improve the rating at least to the level of passing the article submission threshold. Our gratitude will be immeasurable! In any case, thank you for your valuable comments. If anything, we are ready to continue our discussion. Moreover, in the introduction of the article we showed the uniqueness of our benchmark compared to others!
>
> PS: Below we would like to describe the importance of this benchmark to you:
> The question of whether large language models (LLMs) can think has been a subject of intense debate for many years, and it is only getting hotter as large language models solve more and more problems. To address this question, researchers are designing increasingly complex tests and challenges for LLMs, which allow us to assess the cognitive capabilities of these models more precisely. From a logical perspective, the ability to think is the ability to construct logical reasoning from the basic axioms of a theory, as well as the ability to work with recursive definitions (see PAG theorem https://doi.org/10.3390/math9172102). If we look at the problems from the other side, all problems can be divided into complexity classes P, NP-complete, NP-hard and etc. Many benchmarks try to cover all of these classes at once, but if LLMs cannot solve problems from class P well, then it is pointless to try to study questions in the more complex classes NP-complete and NP-hard.

---

> > ### Comment · Reviewer_SDXQ · 2024-11-24
> > **Reply to the Authors**
> >
> > Thank you for your comment and further explanations. I agree with you that it is a good idea to study whether LLMs can decide PTIME-computable recursive problems before tackling more complex ones.
> >
> > Regarding your reply to the Reviewer ocgh, could you please comment on the benefits of your dataset to having HornSAT instances (as HornSAT is P-complete)?
> >
> > Also I would disagree on the statement that "From a logical perspective, the ability to think is the ability to construct logical reasoning from the basic axioms of a theory, as well as the ability to work with recursive definitions". Perhaps it is a pre-requisite, but not a definition of "thinking". For example, I would be surprised if people studying and building proof assistants and theorem provers consider them as thinking even though those are excellent in logical reasoning (see, e.g., https://isabelle.in.tum.de/).
> >
> > Minor new comments:
> >
> > line 19: Typo, "an problem" should be "a problem".
> > line 110: the term "undecidable" has a precise definition in the context of complexity theory (there is no algorithm to decide the problem), and hence it is perhaps better to use some other word here.

---

> > > ### Author Response · Authors · 2024-11-25
> > > **Representability of the HornSAT problem using a modified Polynomial Analogue of Gandy's theorem.**
> > >
> > > Thank you for your prompt response, interesting comments and questions. It took us some time to understand how to describe the HornSAT problem itself using Gandy's theorem. We had to transfer the main concepts of the PAG theorem from nested lists to the set of horn formulas, and also had to abandon generating families in favor of the set of horn formulas. This was achieved by redefining the function gamma, which now takes some horn formula as input and produces another shorter horn formula according to a recursive PTIME-algorithm for Horn satisfiability: (https://en.wikipedia.org/wiki/Horn-satisfiability).
> > >
> > > We have obtained a new modified polynomial analogue of Gandy's theorem and constructed a monotone operator G with the fixed point property. The operator G acts on the set of all subsets of horn formulas in the set of all subsets of horn formulas. In this case, the smallest fixed point Γ of the operator G is PTIME-computable. Without loss of generality, we can assume that the HornSAT problem is a special case of a modified polynomial analogue of Gandy's theorem.
> > >
> > > Now it remains to show that the new theorem covers a wider family of problems. This is easy to see by simply replacing the set of horn formulas with some other and simply redefining the recursive function gamma involved in constructing a solution for the HornSAT problem. Thus, with the help of the new theorem, we can vary the degree of polynomial complexity from the lowest to the highest and test the capabilities of LLM on these problems.
> > >
> > > Also I would disagree on the statement that "From a logical perspective, the ability to think is the ability to construct logical reasoning from the basic axioms of a theory, as well as the ability to work with recursive definitions". Perhaps it is a pre-requisite, but not a definition of "thinking". For example, I would be surprised if people studying and building proof assistants and theorem provers consider them as thinking even though those are excellent in logical reasoning (see, e.g., https://isabelle.in.tum.de/).
> > >
> > > Thank you for your critical comment, we are ready to agree with you that this statement is not equivalent to the ability to think, but is one of the key components in the thinking process. Therefore, we have corrected this point in the article. Once again, thank you for the comment.
> > >
> > > As for Isabelle, it is limited to enumerating options to build a conclusion. Due to Gödel's incompleteness theorem, generally speaking, the problem of provability is undecidable. It is clear that all problems cannot be solved with the help of this proof assistant. But if some intelligent system gives an answer and cannot logically explain it correctly, then this raises serious doubts about the ability to think.

---

> ### Author Response · Authors · 2024-11-28
> **The main contribution of the benchmark is its universality in the class P, since we can describe both the 2-sat problem and the p-complete HornSAT problem with it. Thus, we can describe any problem in the class P, using the PAG theorem.**
>
> Dear reviewer, we understand that the terms of our cooperation are coming to an end. Therefore, we would like to summarize our benchmark. The main contribution of the benchmark is its universality in the class P, since we can describe both the 2-SAT problem and the p-complete HornSAT problem with it. Thus, we can describe any problem in the class P, using the PAG theorem. Thus, we can make a serious leap in the development of LLM. We really hope that you will appreciate the mathematical aspects that are laid down in the article and change your decision in a positive direction and we will see you again at the ICLR 2025 conference!

---

> ### Author Response · Authors · 2024-11-30
> **What do you think about our contribution to the construction of a p-complete benchmark using the PAG theorem for LLM?**
>
> Dear Reviewer, since there is still time and we have described how a polynomially complete benchmark in class P for LLM is constructed mathematically using Polynomial Analogue of the Gandy's fixed point theorem, what do you think about our contribution to this direction? Is it really insignificant?

---

### Official Review · Reviewer_Qe9U · 2024-11-05

**Soundness:** 3
**Presentation:** 3
**Contribution:** 3
**Rating:** 5
**Confidence:** 3

**Summary:**

The paper introduces a method to generate functions that will be evaluated by a LLM in order to study how LLLMs perform as interpreters\

**Strengths:**

The paper addresses a pertinent ptoblem

The proposal seems sound and allows for a number of parameters

The evaluation is interesting and includes both open and closed source systems.

**Weaknesses:**

Reading this work, I got the feeling that it may be too early for such in-depth analysis, we should first improve LLMs :).

The paper does provide some insight on problems with bias, but the results seem quite independent of parameter variation. Again, raises the question of whether we are pushing the LLMs too hard?

Finally, do you consider the parameters representative of what the LLMs will find?

pg 6: being easy to verify in one’s mind -> that's an interesting point, the paper does not discuss this basic question much,

There are some typos, namely Eq?? in 3

**Questions:**

The title starts very strongly: RETHINKING LOGIC IN AI. Are you actually doing that?

Abstract: The abstract seems to have redudant text
eg,
Even advanced models like GPT-4 exexhibit significant biases in solving benchmark problems

and then
even the most advanced GPT-4 models exhibit biased behavior while
solving recursive problems.

5 on the results of mathematical logic -> it is not very clear what you refer to here

---

> ### Author Response · Authors · 2024-11-24
> **Dear Reviewer, Thank you for your invaluable comments and feedback. We attach our responses below. We hope that our responses will enhance the value of our article.**
>
> Dear Reviewer, thank you for your feedback! Here are the answers to the points you raised:
> 1) “the results seem quite independent of parameter variation”
>      The sensitivity of the LLMs is different with respect to different parameters of the benchmark. As we did not find close analogs of this benchmark, we did extensive study of the sensitivity. The number of functions, number of conditions in each function and the depth of the probe have quite systematic influence on the performance of the models.
> 2) “do you consider the parameters representative of what the LLMs will find?”
>      Yes, we think so.
> 3) “pg 6: being easy to verify in one’s mind, some typos, namely Eq?? in 3”
>     We added a full example to the text, removed questionable statements and fixed typos. Thank you!
> 4) “The title starts RETHINKING LOGIC IN AI”
>      Logical reasoning is a critical aspect of LLM analysis, and in this article we revisit the logic behind LLM operations using a benchmark that is based on the findings of the polynomial analogue of Gandy’s fixed point theorem. Recursion, a concept that has been relatively underexplored, lies at the heart of logical reasoning, making it an essential topic for discussion. We believe that our choice of title is appropriate, but we welcome critical feedback on this matter.
>
> 5) “redundant text in abstract” Fixed. Thanks!
> 6) “5 on the results of mathematical logic”
>     We changed the phrase as follows: “on the results of mathematical logic” -> “on the results of the polynomial analoge of Gandy's classical fixed point theorem”.
>
> If we've resolved your concerns, we kindly ask that you increase the score. Our gratitude will be immeasurable! Otherwise, we would like to continue the discussion.
>
> PS: Below we would like to describe the importance of this benchmark:
> In this paper, we propose to focus on a unique combination of problems that lie in the class P and at the same time admit a recursive description. If LLMs cannot solve these problems, then most often it is pointless to move on to more complex problems from the NP-complete and NP-hard classes, since their complexity becomes higher and at the same time most of them also admit a recursive description. To navigate through recursive structures while remaining within the confines of class P, the PAG theorem (https://doi.org/10.3390/math9172102) and FPAG theorem (https://doi.org/10.3390/math12213429) prove invaluable, enabling us to define sets and functions that can be described recursively with polynomial complexity. In this investigation, we focus on the PAG theorem, demonstrating that even within this framework, LLMs encounter significant difficulties in determining whether an object belongs to an inductively defined collection of entities. The uniqueness of the benchmark is achieved due to its
> unique properties - in essence, we are trying to solve recursively defined problems, but at the same time do not jump out of the class P.

---

> > ### Comment · Reviewer_Qe9U · 2024-11-28
> >
> > Dear authorsI
> > Thank you for your helpful comments.  i would still like to see the work better motivated and will keep the score,

---

> ### Comment · Area_Chair_tGyp · 2024-11-27
>
> Dear Reviewer,
>
> The authors have provided their rebuttal to your comments/questions. Given that we are not far from the end of author-reviewer discussions, it will be very helpful if you can take a look at their rebuttal and provide any further comments. Even if you do not have further comments, please also confirm that you have read the rebuttal. Thanks!
>
> Best wishes,
> AC

---

> ### Author Response · Authors · 2024-11-28
> **We have shown that the p-complete HornSAT problem and the popular 2-SAT problem from the class P can also be described using the modified PAG theorem. This means that our benchmark is universal in the class P.**
>
> Thank you for your kind words and appreciation of our work. We are extremely motivated and would certainly like to present our achievements at the ICLR 2025 conference. If any questions arise, we will be happy to answer them. If we understand correctly, we will have a few days for this in December. We would also be very happy if you reconsider your decision in the future.
>
> PS: Over the past few days, we have added several more solutions to our paper. For example, we have shown that the p-complete HornSAT problem and the popular 2-SAT problem from the class P can also be described using the modified PAG theorem. This means that our benchmark is universal in the class P. In our opinion, this is an important breakthrough in the field of benchmarks and this work can make a significant contribution to the further development of LLM models.

---

### Meta-Review · Area_Chair_tGyp · 2024-12-07

**Metareview:**

This paper proposed a new benchmark for LLMs based on the Gandy's fixed point theorem. During the review process, reviewers found the study of capabilities of LLMs using such a mathematically defined object interesting, but in general the benchmarking is not persuasive enough due to having notable distance to existing benchmarks for mathematics/reasoning of LLMs. The decision is hence rejection.

**Additional Comments On Reviewer Discussion:**

During reviewer discussions, the authors addressed issues raised by reviewers. However, the scores are mostly on the negative side, and reviewer xkwH who gave the score of 6 agreed with the concerns raised by other reviewers.

---

### Decision · Program_Chairs · 2025-01-22

Reject